# Contribution of Even/Odd Sound Wave Modes in Human Cochlear Model on Excitation of Traveling Waves and Determination of Cochlear Input Impedance

Wenjia Hong [1] and Yasushi Horii [2],*

1   Graduate School of Informatics, Kansai University, Takatsuki 569-1095, Japan; hwjh2654568@gmail.com
2   Faculty of Informatics, Kansai University, Takatsuki 569-1095, Japan
*   Correspondence: horii@kansa-u.ac.jp; Tel.: +81-(72)690-2476

**Abstract:** Based on the Navier–Stokes equation for compressible media, this work studies the acoustic properties of a human cochlear model, in which the scala vestibuli and scala tympani are filled with compressible perilymph. Since the sound waves propagate as a compression wave in perilymph, this model can precisely handle the wave–based phenomena. Time domain analysis showed that a sound wave (fast wave) first propagates in the scala vestibuli and scala tympani, and then, a traveling wave (slow wave) is generated by the sound wave with some delay. Detailed studies based on even and odd mode analysis indicate that an odd mode sound wave, that is, the difference in the sound pressures between the scala vestibuli and scala tympani, excites the Békésy's traveling wave, while an even mode sound determines the input impedance of the cochlea.

**Keywords:** auditory mechanism; cochlea; traveling wave theory; compressible perilymph; even/odd mode analysis

## 1. Introduction

According to Békésy's traveling wave theory [1], the basilar membrane (BM) in the cochlea plays an important role in analyzing the frequency spectrum of sound waves. The BM has a long trapezoid shape, and becomes narrower and thicker near the base of the cochlea. In approaching the apex, it becomes wider and thinner, and increases in flexibility. As a result, higher frequency sounds cause a displacement of the BM near the base, and lower frequency sounds cause a displacement near the apex. These displacements are detected by the outer hair cells, which are regularly arranged on the BM [2]. By using such an auditory system, humans hear sounds in an audible frequency range from 20 Hz to 20,000 Hz and in a huge dynamic range of about 120 dB [3]. Even after Békésy's proposal of the traveling wave theory, considerable progress and great discoveries have continued, including the active process of a cochlear amplifier [4]. However, there are still many questions regarding the auditory mechanism.

In looking at the cochlear models reported in recent years, the finite element method (FEM) based on fluid dynamics is primarily used due to its design flexibility for three–dimensional cochlea models [5–13]. In fact, some models include the scala vestibuli (SV), scala tympani (ST), BM, oval window (OW), round window (RW), and even the cochlear aqueduct. These models can be classified using the following two aspects. One is the shape of the cochlea models, that is, a simplified straight–type cochlear model and a realistic spiral–shaped cochlear model. Though there might be a small difference between their simulation results, both models are considered to be useful to study the basic mechanisms of the cochlea. The other aspect is the settings of the perilymph, which fills the SV and ST. When the perilymph is set as a compressible medium, compression waves (that is, sound waves) are allowed to propagate in the SV and ST, and their wavelengths and phases are precisely processed in the cochlear simulation. However, if the perilymph is set as

an incompressible medium, the perilymph is not compressed at all, and the wavelength in the medium would be infinity. Though such an approximation might be useful at a lower audible frequency, errors can be more significant at a higher frequency.

This paper designs a cochlear model assuming that the perilymph in the SV and ST is compressible, and explains the excitation mechanism of a traveling wave based on even and odd mode analysis [14]. This is a basic technique that has been used for a long time in microwave engineering for designing parallel couplers and explaining the principle of undesired crosstalk on analog phones. We explain this technique briefly in Section 3. By applying the even and odd mode analysis technique to FEM–based simulations, we explain that an even sound wave mode and an odd sound wave mode are excited simultaneously in the cochlea, and the odd mode contributes to generating Békésy's traveling waves on the BM and the even mode forms a standing wave in the cochlea. Though these modes show different behaviors, both modes cannot exist independently and are deeply related to each other so as to meet the boundary conditions of the cochlea, such as the reflection conditions at the round window and the apex of the cochlea. In addition to this brand–new approach, we also develop a new cochlear–equivalent circuit, based on the transmission line theory [14]. From these simulation results, we conclude that the input impedance of the cochlea is defined by the parallel connection of the input impedance of the even mode and the odd mode, which means that the overall performance of the cochlea is determined by a combination of the even and odd mode sound waves. Finally, we would like to emphasize that it is indispensable to define the perilymph as a compressible medium to discuss the even and odd mode approach.

## 2. Compressible Perilymph–Filled Cochlea Model

### 2.1. Modeling

The spiral shaped human cochlea, which is housed in a hard temporal bone, consists of the SV, scala media (SM), and ST. The SV and SM are separated by the Reissner membrane, and the SM and ST are separated by the BM. In the SM, there is an organ of Corti containing outer hair cells (OHCs) and inner hair cells (IHCs), which detect the spectrum of sound waves and transmit sound information to the auditory cortex through auditory nerves.

In the model, the microscopic behavior of the OHCs and IHCs, including the functions of the cochlea amplifier, were ignored, and the influence of the thin Reissner's membrane was neglected because these effects are considered to be negligible and limited when we discuss how a sound wave is generated and travels in the cochlea from a macroscopic point of view [15]. In addition, the spiral shaped cochlea was unrolled to simplify the model structure, and only the important portions that determine wave propagation phenomena remained. In particular, the perilymph of the SV and ST was assumed as a compressible fluid in order to precisely treat sound wave propagation as a compression wave.

Figure 1 shows a straight–shaped cochlea model, including the compressible perilymph. The model was configured using three layered blocks of the SV, SM, and ST. The fan–shaped BM, made of an elastic material, was embedded in the SM, and a small helicotrema hole, which connects the SV to the ST, was on the apical side of the SM. An ideal hard wall condition, which provides 100% reflection with a maximum pressure level and zero fluid velocity, was applied to the walls of the SV and ST, except for the portions facing the oval window (OW), round window (RW), and BM. The OW plane was set as an input port, and a sinusoidal sound wave was excited there. An ideal soft wall condition, which provides a 100% reflection with a zero pressure level and maximum fluid velocity, was applied to the RW to express the reflection of a sound wave coming from the perilymph–filled ST ($Z_0 = 1.5$ MPa s/m$^3$) to an air–filled middle ear cavity ($Z_0 = 440$ Pa s/m$^3$), where $Z_0$ is the characteristic acoustic impedance of the medium. The SV, ST, and helicotrema were filled with compressible perilymph with a viscosity of 0.7027 mPa s, a velocity of 1520 m/s, and a mass density of 994.6 kg/m$^3$. A Young's modulus of 1 MPa, Poisson's ratio of 0.49, and a mass density of 1200 kg/m$^3$ were used for the BM [16]. Other structural parameters are given in the caption of Figure 1.

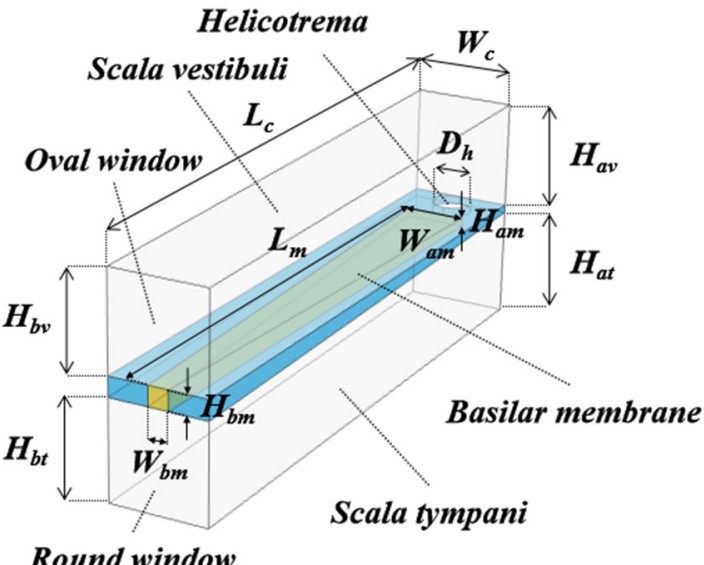

**Figure 1.** A straight–shaped cochlear model. The SV, ST and helicotrema are filled with a compressible perilymph. The fan–shaped BM is embedded in the SM. $L_c$ = 35 mm, $W_c$ = 1.2 mm, $H_{bv} = H_{bt}$ = 1.235 mm, $H_{av} = H_{at}$ = 1.245 mm, $W_{bm}$ = 100 μm, $H_{bm}$ = 30 μm, $W_{am}$ = 500 μm, and $H_{am}$ = 10 μm [17–20].

A time domain simulation was carried out using the commercial software COMSOL Multiphysics Ver. 5.3, including an add–on acoustic module and structural mechanics module. For the simulations, the continuity equation and Navier–Stokes equation for compressible media were used:

$$\frac{\partial \rho}{\partial t} + \nabla \cdot (\rho \mathbf{u}) = 0 \tag{1}$$

$$\rho \left( \frac{\partial \mathbf{u}}{\partial t} + (\mathbf{u} \cdot \nabla)\mathbf{u} \right) = -\nabla p + \nabla \cdot (\mu(\nabla \mathbf{u} + (\nabla \mathbf{u})^T) - \frac{2}{3}\mu(\nabla \cdot \mathbf{u})\mathbf{I} \tag{2}$$

where $\mathbf{u}$ stands for the fluid velocity, $\mathbf{I}$ is the identity tensor, $p$ is the fluid pressure, $\rho$ is the fluid density, and $\mu$ is the viscosity.

*2.2. Frequency Domain Analysis*

Frequency domain analysis was carried out for the cochlear model mentioned above. A 1 Pa sinusoidal sound wave was excited at the OW plane, and the sound pressure level (SPL) of the SV (blue) and ST (green) and the displacement of the BM (red) were simulated at 1000 Hz, 5000 Hz, 9000 Hz, 13,000 Hz, and 17,000 Hz, respectively. The SPLs were calculated on the observation lines that ere set just at the center of the SV and ST blocks; results are shown in Figure 2a–e. The horizontal axis of each graph shows the internal position of the cochlea. For example, the positions of 0 mm and 35 mm correspond to the cochlear base and apex, respectively. Looking at the time domain responses, it can be confirmed that the SPL of the SV varied, up and down, across zero pressure at the cochlear base, while the SPL of the ST became zero at the same time due to the free–end reflection condition (reflection with a zero pressure level and maximum fluid velocity) of the RW. Especially, it should be noted that the SPLs of the SV and ST partially vibrated near the base; however, this vibration disappeared at the apex side beyond the yellow dashed lines. Instead, the displacement of the BM occurred, and the traveling wave began to travel on the BM.

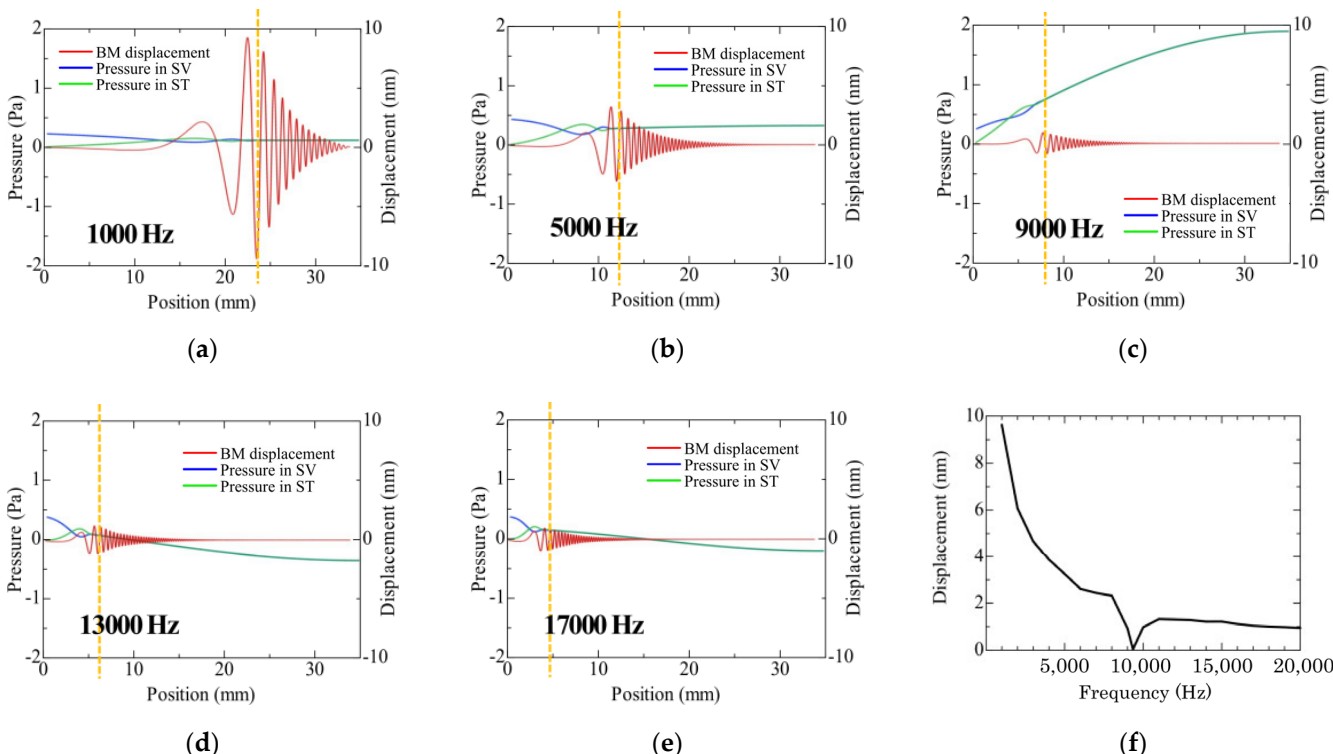

**Figure 2.** (**a**–**e**) Frequency domain analyses of the sound pressure level (SPL) of the SV (blue) and ST (green), and the displacement of the BM (red) when the OW plane is excited by a 1 Pa sinusoidal sound wave. The horizontal axis shows the internal position of the cochlea. (**f**) Frequency characteristic of the maximum displacement of the BM, extracted from the maximum displacements of the BM such as those in (**a**–**e**).

The curves drawn in red in Figure 2a,b show the displacement of the BM at a certain time. By checking the envelope of the displacement as a movie, the maximum displacement of the BM and the position in the cochlea were then determined. Taking the responses at 1000 Hz and 5000 Hz, for example, the maximum displacements of the BM were 9.63 nm at 1000 Hz and 3.25 nm at 5000 Hz, respectively. We repeated this procedure for other frequencies, and finally obtained the frequency dependence of the maximum displacement of the BM, as shown in Figure 2f. In addition, we confirmed that the position where maximum displacement occurred on the BM was coincident with Greenwood's cochlear tonotopy [21]. This graph indicates that the displacement tends to be larger at lower frequencies. Generally, when an elastic membrane is excited by a sinusoidal sound wave with the same pressure level, the displacement will be larger when the sound frequency is lower. The behavior of such an elastic membrane can be seen in our simulation results as well. Of course, these results do not include the non–linearity and amplification effects of the cochlear amplifier; however, the response can be considered as the sensitivity of the cochlea to the initial sound stimuli.

### 2.3. Time Domain Analysis

A 5000 Hz sound wave with a 1 Pa sound pressure was set for the OW plane, and time domain analyses were carried out. Figure 3 shows the side view of the straight–shaped cochlea. An upper and a lower clock correspond to the SV and ST, respectively. Since the SM is thin compared to the thickness of the SV and ST, the SM is not clearly shown, but exists between the SV and ST. The D–BM color bar in the figure shows the displacement of the BM. The SPL color bar shows the sound pressure levels in the SV and SM.

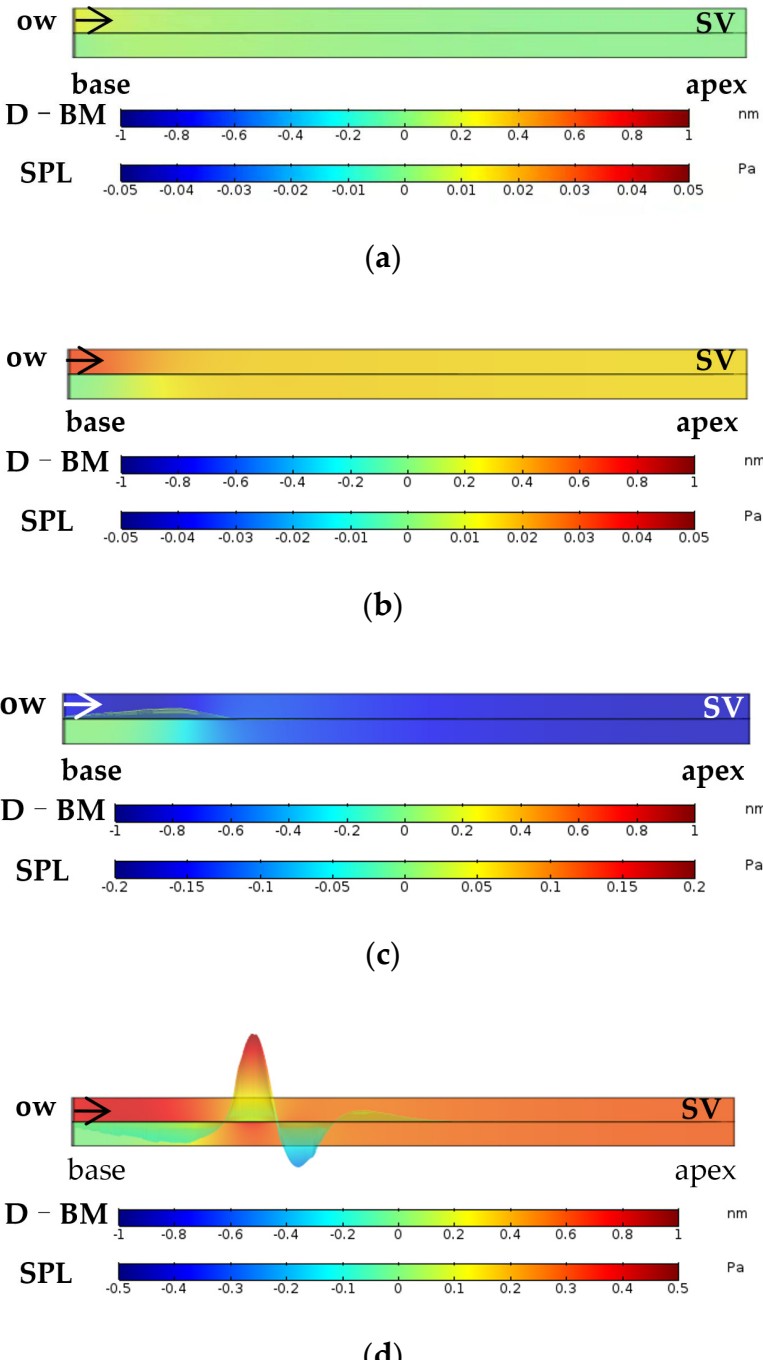

**Figure 3.** Time domain analysis of the sound pressure level (SPL) of the SV and ST and the displacement of the BM (D–BM) when the OW is excited by a 1 Pa, 5000 Hz pure tone. (**a**) *t* = 0.03 ms, (**b**) *t* = 0.06 ms, (**c**) *t* = 0.17 ms, and (**d**) *t* = 0.46 ms.

As shown in Figure 3a, at *t* = 0.03 ms, the excitation had only just started, and the SPL was still almost zero at any point in the SV and ST. However, as shown in Figure 3b, at *t* = 0.06 ms, the sound wave traveled as a fast wave with a speed of 1520 m/s in the perilymph, and the SPL became zero only near the RW region. This is due to the free–end reflection condition of the RW. At this time, the traveling wave was not yet seen.

When the time reached *t* = 0.17 ms, the traveling wave began to appear near the base of the cochlea, as shown in Figure 3c. It should be noted that the SPL near the RW was still kept at a zero–pressure level, but, near the apex of the cochlea, the SPLs in the SV and ST increased and were almost identical.

As presented in Figure 3d, when time reached $t = 0.46$ ms, the traveling wave was gradually formed on the BM. Although the wavelength of the vibration waveform observed on the BM looked shorter than that of the 5000 Hz sound wave (fast wave with a wavelength of 304 mm in the perilymph), the complete Békésy's traveling wave (slow wave) was not formed yet. To form an actual traveling wave in which the wavelength is significantly shortened and compressed on the BM, it will take dozens of milliseconds to reach the steady state.

The upper graphs of Figure 4 present the SPLs in the SV and ST and the displacement of the BM after reaching the steady state ($t = 38.94$ ms), when the OW was excited by a 1 Pa, 5000 Hz pure tone. The horizontal axis shows the internal position of the cochlea. Some interesting results were found in the simulation, as follows.

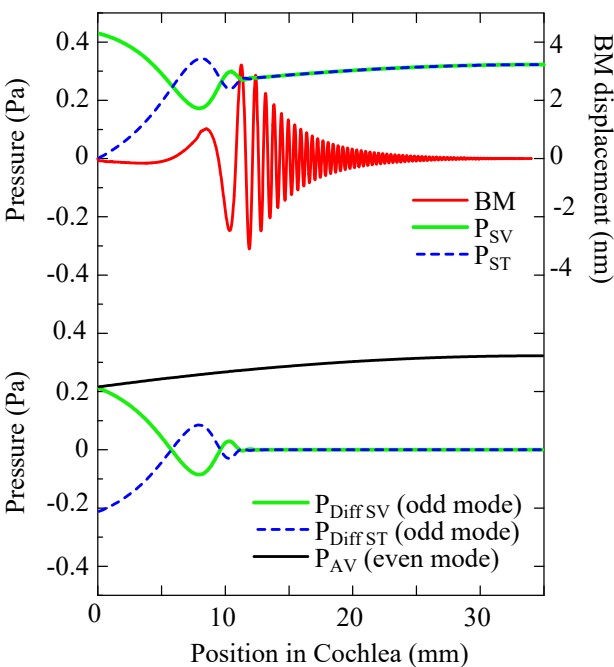

**Figure 4.** The upper graphs show the sound pressure levels (SPLs) in the SV ($P_{SV}$ in green) and ST ($P_{ST}$ in blue) and the displacement of the BM (BM in red) when a 1 Pa, 5000 Hz pure tone is applied to the OW. The lower graphs present the SPLs of the even symmetric sound mode ($P_{AV}$ in black) in the SV and ST, and odd symmetric sound mode in the SV ($P_{Diff\_SV}$ in green) and ST ($P_{Diff\_ST}$ in blue). The horizontal axis shows the internal position of the cochlea, and all graphs reach the steady state ($t = 38.94$ ms).

1. The SPL of the ST became zero all the time at the base of the cochlea (at 0 mm position) due to the free–end reflection condition of the RW.
2. As shown by the SPLs of the SV and ST, a small vibration can be seen in each graph at the position from 0 mm to 12 mm.
3. The SPLs of the SV and ST, corresponding to the graphs of the $P_{SV}$ and $P_{ST}$, completely overlapped from 12 mm to 35 mm.
4. The displacement of the BM became the maximum at around 12 mm, at the position where the tonotopy predicts that humans hear 5000 Hz sounds.

Here, we calculated the average pressure, $P_{AV} = (P_{SV} + P_{ST})/2$, and the pressure difference, $P_{Diff\_SV} = P_{SV} - P_{AV}$ and $P_{Diff\_ST} = P_{ST} - P_{AV}$. The results are shown in the lower graphs in Figure 4. Interestingly, the graphs ($P_{Diff\_SV}$ and $P_{Diff\_ST}$) show perfect symmetric against the zero axis and $P_{AV}$ draws a smoothly changing waveform. In other words, this means that the sound wave traveling in the cochlea is composed of an even symmetric sound wave (even mode) and an odd symmetric sound wave (odd mode).

### 3. Even/Odd Mode Analysis

The cochlea model shown in Figure 1 has an architecture in which the SM is symmetrically sandwiched by the same–shaped SV and ST. This means that even and odd mode analyses can be applied to the model. Even and odd mode analyses are basic techniques that have been used for a long time in microwave engineering for designing couplers using parallel transmission lines and for explaining the principle of the crosstalk often seen in analog telephones [14]. As shown in Figure 5a, two transmission lines, A and B, were placed in parallel, and the terminals were numbered from Port 1 to Port 4. Then, when a signal was input from Port 1, it was transmitted along transmission line A and output from Port 3. However, depending on the degree of coupling between the lines, the signal in transmission line A leaked somewhat into transmission line B. This problem is expressed by the sum of two transmission line modes. One is an even symmetric mode (even mode) that was generated by exciting Port 1 and Port 2 with the same amplitude and in–phase signals, as shown in Figure 5b. The other one is an odd symmetric mode (odd mode) that was generated by exciting Port 1 and Port 2 with the same amplitude, but with anti–phase signals, as shown in Figure 5c. Simply through the superposition of these modes, the excitation at Port 1 in Figure 5a could be treated as the sum of the even and odd mode excitations [14].

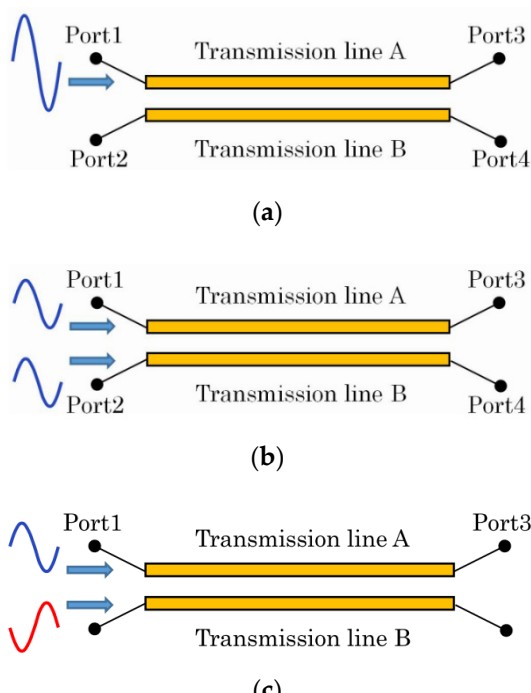

**Figure 5.** A coupler composed of two transmission lines. (**a**) Geometry and Port 1 excitation. (**b**) Port 1 and Port 2 excitation with the in–phase signals (even mode). (**c**) Port 1 and Port 2 excitation with the anti–phase signals (odd mode).

In this section, in order to evaluate how the even and odd modes of the sound wave (fast wave) contribute to generating the traveling wave (slow wave), a symmetric model (Figure 6) was designed based on the cochlear architecture shown in Figure 1. This model has two input planes, designated as "input 1" and "input 2" in the figure. In the even mode, these planes were excited by the same in–phase amplitude sound waves, while in the odd mode, they were excited by the same anti–phase amplitude sound waves. To avoid undesired coupling between the odd mode sound waves in the SV and ST, the helicotrema was removed from the original cochlear model, and only the BM remained as it was. Other physical and structural parameters were completely same as the original model in Figure 1.

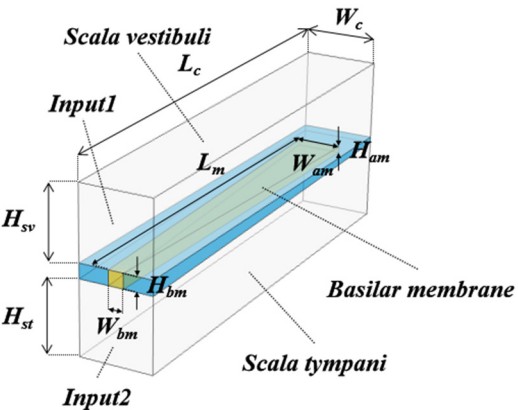

**Figure 6.** A cochlea–based symmetric model for even and odd mode analyses. Two input planes "input 1" and "input 2" are excited by in–phase (even mode) and anti–phase (odd mode) sound waves. The helicotrema is removed from the original model. Other structural parameters are completely the same as those of the cochlea model in Figure 1.

The simulation results are summarized in Figure 7 when a 1 Pa, 5000 Hz sound wave was applied for both input planes. The SPLs in the SV and ST and the displacement of the BM, which was generated by the even or odd symmetric sound waves, are presented. As shown in Figure 7a, in the odd mode, odd symmetric SPLs could be observed in the SV and ST at the internal cochlear position from 0 mm to 12 mm. However, the odd mode response disappeared beyond 12 mm. This means that the odd mode sound wave plays an important role in generating the traveling wave. The color bar also presents this phenomenon visually. It should be noted that the energy of the odd mode sound wave was perfectly transformed into that of the traveling wave on the BM.

On the other hand, in the even mode, as shown in Figure 7b, the SPL graphs of the SV and ST completely overlapped with each other and formed an even symmetric sound wave in the cochlea. However, the displacement of the BM was not seen at all. This can be confirmed visually with the color bar. This means that the even mode sound wave does not generate the traveling wave. Though only the demonstration result at 5000 Hz is shown here, the same can be said for other frequencies.

The results mentioned above are summarized in Figure 8a; namely, the sound wave excited at the OW generated two symmetric sound modes, an even mode and an odd mode. Focusing on the wave phenomena in the SV, the even mode traveled in the SV, and was reflected at the cochlear apex with the maximum sound pressure level and zero fluid velocity, and finally, it came back to the OW without power dissipation, while in the odd mode, since the sound wave was completely transformed into the traveling wave and absorbed in the BM, the odd mode sound did not come back to the OW. These phenomena can be simply expressed by an equivalent circuit based on transmission line theory, as shown in Figure 8b.

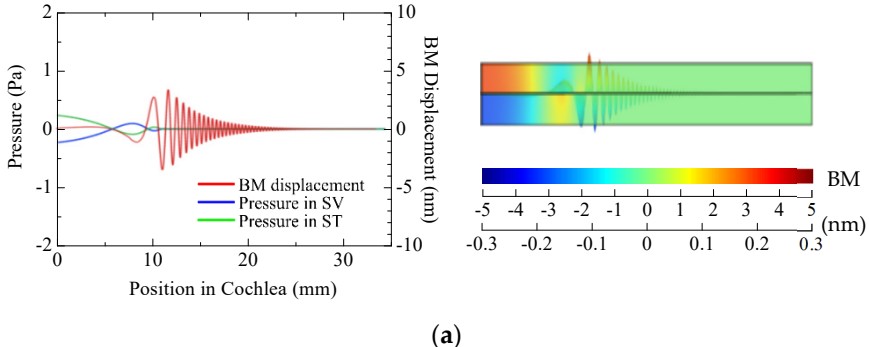

(**a**)

**Figure 7.** *Cont.*

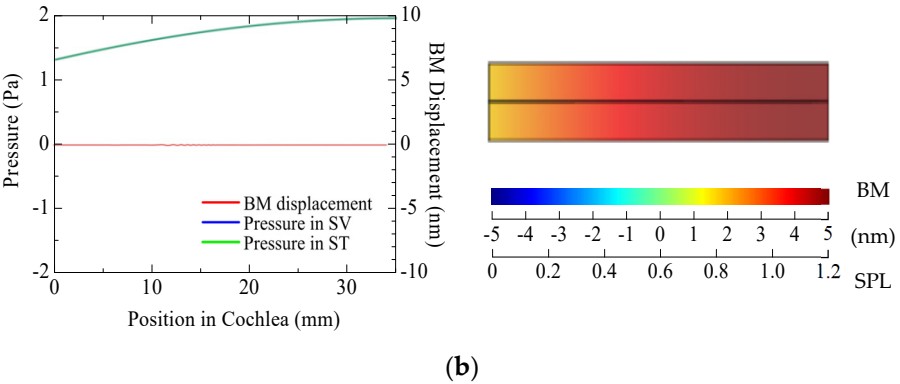

(**b**)

**Figure 7.** Even and odd mode simulation results for the cochlea–based symmetric model presented in Figure 6. The horizontal axis shows the internal position of the cochlea. (**a**) Odd mode and (**b**) even mode.

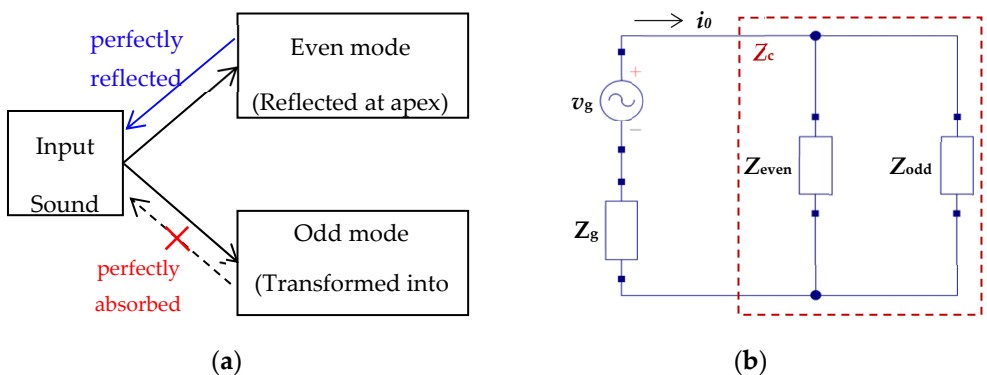

(**a**)                                        (**b**)

**Figure 8.** An equivalent circuit of the cochlea. (**a**) Contributions of the even and odd sound wave modes to the cochlear acoustics. (**b**) an equivalent circuit of the cochlea designed based on transmission line theory.

In this circuit, an alternative voltage source, $v_g$, and the inner impedance of the source, $Z_g$, were connected in series to design an excitation circuit, and the input impedance of the cochlea, $Z_c$, wa expressed by a parallel connection of an even mode impedance, $Z_{even}$, and an odd mode impedance, $Z_{odd}$. Since the even and odd modes behave independently in the SV and ST, the coupling of both modes was ignored. Then, $Z_{even}$ and $Z_{odd}$ were written as (see Appendix A):

$$Z_{even} = -jZ_0 \cot \beta L_c \tag{3}$$

$$Z_{odd} = Z_0 \tag{4}$$

$$\beta = \omega \sqrt{\frac{\rho}{\kappa}} \tag{5}$$

where $j$ is the imaginary unit ($j^2 = -1$), $Z_0$ is the acoustic characteristic impedance of the perilymph, $\rho$ is the mass density, $\kappa$ is the bulk modulus, $f$ is the sound wave frequency, $\beta$ is the phase constant, and $L_c$ is the cochlear length, respectively.

From these, the total input impedance of the cochlea $Z_c$ was expressed by the parallel connection of the $Z_{even}$ and $Z_{odd}$ as follows:

$$Z_c = \frac{Z_{even} Z_{odd}}{Z_{even} + Z_{odd}} \tag{6}$$

Circuit current $i_0$ was calculated as:

$$i_0 = \frac{v_g}{Z_g + Z_c} \tag{7}$$

Then, the applied voltage ($v_c$) to the $Z_{even}$ and $Z_{odd}$ were estimated by:

$$v_c = i_0 Z_c \tag{8}$$

and currents on the $Z_{even}$ and $Z_{odd}$ were given by:

$$i_{even} = \frac{v_c}{Z_{even}} \tag{9}$$

$$i_{odd} = \frac{v_c}{Z_{odd}} \tag{10}$$

Finally, the power dissipation on the even and odd modes, $PD_{even}$ and $PD_{odd}$, and the total power dissipation by the cochlea, $PD_c$, were expressed as:

$$PD_{even} = \frac{v_c i_{even}^*}{2} \tag{11}$$

$$PD_{odd} = \frac{v_c i_{odd}^*}{2} \tag{12}$$

$$PD_c = PD_{even} + PD_{odd} = \frac{v_c i_0^*}{2} \tag{13}$$

In the circuit simulations, parameters $v_g$ = 1.0 V (corresponding to a 1 Pa sound wave excitation), $Z_0$ = 1.5 MPa s/m$^3$, $\rho$ = 994.6 kg/m$^3$, $\kappa$ = 2.30e9 Pa, $f$ = 5000 Hz, and $L_c$ = 35 mm were used.

The circuit simulation results are shown in Figure 9. The frequency dependence of the even and odd mode impedance, $Z_{even}$ and $Z_{odd}$, are shown in Figure 9a,b. Blue solid curves and red dashed curves denote the real part and imaginary part of the impedance, respectively. As shown by $Z_{even}$, the real part of $Z_{even}$ became zero for all frequencies because the even mode sound wave was perfectly reflected at the apex and went back to the voltage source again without power dissipation. On the other hand, at lower frequencies, below 10,600 Hz, the imaginary part of the $Z_{even}$ became negative, which meant that the cochlea became capacitive. Contrarily, at higher frequencies beyond 10,600 Hz, it became positive, and the cochlea became inductive. This result tells us an important fact in that the $Z_{even}$ reached zero at around 10,600 Hz. In this case, since $Z_{even}$ was connected to $Z_{odd}$ in parallel, as shown in Figure 8b, the input impedance of the cochlea, $Z_c$, also reached zero, and sound stimuli could not enter the cochlea. As a result, hearing loss will occur at this frequency. In fact, the displacement of the BM shown in Figure 2f was also confined around there. Furthermore, note that hearing deterioration is confirmed at the measured equal–loudness contours around 10,000 Hz [22]. We need to compare both results carefully because the equal loudness contours include psychological effects on the test subjects. However, our simulation result implies that the hearing deterioration at this frequency might occur due to the structure of the cochlea.

Next, the frequency dependence of the even and odd mode power dissipation, $PD_{even}$ and $PD_{odd}$, are presented in Figure 9c,d. The graph of the $PD_{even}$ indicates that only the reactive power was stored in the cochlea and there was no power dissipation at any frequency. In contrast, the graph of the $PD_{odd}$ shows that the cochlea dissipated active power to displace the BM and generate the traveling waves. However, the power dissipation was reduced around 10,600 Hz, because the even mode impedance, $Z_{even}$ = 0, affected the odd mode properties.

The input impedance of the cochlea, $Z_c$, is expressed by Equation (6), and the frequency dependence of $Z_c$ is summarized in Figure 9e. As pointed out, the zero impedance of the even mode $Z_{even}$ = 0 at 10,600 Hz affected the whole hearing performance of the cochlea; namely, the input impedance of the cochlea also became zero, and a significant impedance mismatch between the source and the cochlea occurred. As a result, although the cochlea as excited by the 1 Pa ($v_g$ = 1.0 V) sound source, the actual pressure applied to the cochlea

became smaller. The red point shows an applied pressure at 5000 Hz, calculated by the FEM–based structural simulation of the even and odd mode sound waves in Figure 4. Although there are some errors between the results obtained by the structural analysis and the equivalent circuit analysis, we consider that they are within an acceptable range.

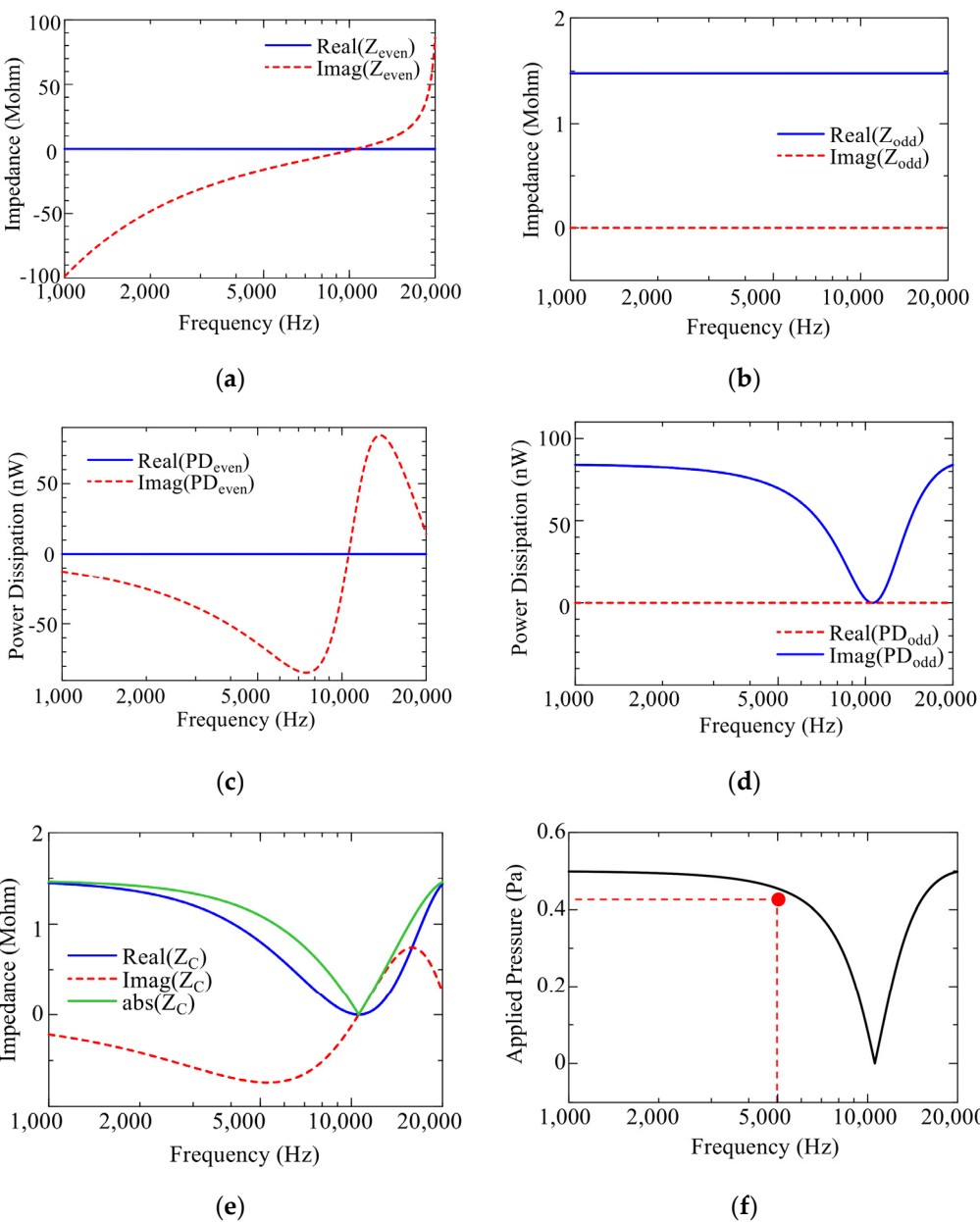

**Figure 9.** Equivalent circuit simulation results when a 1 Pa input ($v_g$ = 1.0 V) is applied to the voltage source in Figure 8b. (**a**) Input impedance of the even mode. (**b**) Input impedance of the odd mode. (**c**) Power dissipation of the even mode. (**d**) Power dissipation of the even mode. (**e**) Input impedance of the cochlea. (**f**) The actual pressure applied to the cochlea under the impedance mismatch between the source and cochlea. The red point shows an applied pressure at 5000 Hz, which is calculated by the FEM–based structural simulation of the even and odd mode sound waves in Figure 4.

## 4. Conclusions

The wavelengths of the sound wave propagating in the perilymph at a speed of 1520 m/s were estimated as $\lambda$ = 15.2 m at 100 Hz and $\lambda$ = 1.52 m at 1000 Hz, respectively. These wavelengths were long enough compared to the size of the cochlea. However, at 10,000 Hz, the wavelength was only $\lambda$ = 152 mm, and the 35 mm cochlea corresponded

to about a quarter wavelength of the sound wave. Since the total length of the SV and ST was about 70 mm, the OW and RW were located at a half wavelength distance at this frequency. This means that, when the OW was pushed in, the RW moved toward the inside simultaneously. In reality, the physics of the cochlea are not so easy to understand because it includes elastic media such as the BM, OW, and RW. However, considering the wavelength of the sound waves at the higher audible frequency, we need to define the perilymph as a compressible medium to treat the phenomena caused by the sound waves in the cochlea more precisely.

Due to these reasons, we treated the perilymph of the SV and ST as a compressible medium, and fluid dynamics simulations were carried out using the Navier–Stokes equation for compressible media. From the time domain simulations, we observed that the sound wave (first wave) started to propagate in the cochlea first, and then the traveling wave (slow wave) as generated on the BM after a delay. This indicates that sound waves play an important role in exciting the traveling wave in the cochlea. From the frequency domain simulations, we observed that (1) the odd mode sound waves contributed to exciting the traveling wave; (2) the even mode sound waves did not generate the traveling wave; and (3) both modes did not couple in the cochlea. However, as indicated by the equivalent circuit simulations based on the transmission line theory, the even and odd modes are deeply related in terms of the impedance of the cochlea. For example, the cochlea is expected to lose hearing capability at 10,600 Hz due to the zero impedance of the even mode. Finally, we hope that many new discoveries will be made by assuming the compressible perilymph when the acoustic aspects of the cochlea are studied.

**Author Contributions:** Conceptualization, W.H. and Y.H.; software, W.H.; writing—original draft preparation, W.H.; writing—review and editing, Y.H.; supervision, Y.H.; project administration, Y.H.; funding acquisition, Y.H. All authors have read and agreed to the published version of the manuscript.

**Funding:** This research was financially supported by the Kansai University Fund for the Promotion and Enhancement of Education and Research, 2020. "Engineering study on auditory mechanism for innovative development of clinical medicine".

**Institutional Review Board Statement:** Not applicable.

**Informed Consent Statement:** Not applicable.

**Data Availability Statement:** The data that support the findings of this study are available from the corresponding author upon reasonable request.

**Conflicts of Interest:** No conflict of interest.

## Appendix A

Under the assumption that a fluid is thermodynamically uniform and constant, the basic equation of fluid dynamics was linearized by adding small disturbance terms to velocity, $v$; math density, $\rho$; and pressure, $p$, and ignoring the terms of degree 2 and higher. Then, the wave equation in terms of sound waves was derived as follows:

$$\frac{\partial^2 p}{\partial t^2} - c^2 \left( \frac{\partial^2 p}{\partial x^2} + \frac{\partial^2 p}{\partial y^2} + \frac{\partial^2 p}{\partial z^2} \right) = 0 \tag{A1}$$

By assuming that the sound wave travels in $x$ direction, this equation was simplified as:

$$\frac{\partial^2 p(x)}{\partial t^2} - c^2 \frac{\partial^2 p(x)}{\partial x^2} = 0 \tag{A2}$$

$$c = \sqrt{\frac{\kappa}{\rho}} \tag{A3}$$

where $\kappa$ is the bulk modulus, $\rho$ is the mass density, and $c$ is the speed of the sound wave. When the sound source was expressed by:

$$p_0(t) = P_0 e^{j\omega t} \tag{A4}$$

Equation (A2) could be written as:

$$\frac{\partial^2 p(x)}{\partial x^2} + \beta^2 p(x) = 0 \tag{A5}$$

where

$$\beta = \frac{\omega}{c} = \omega \sqrt{\frac{\rho}{\kappa}} \tag{A6}$$

In addition, the velocity of fluid $u$ was given by:

$$\frac{\partial u(x)}{\partial t} = j\omega u(x) = -\frac{1}{\rho_0} \frac{\partial p(x)}{\partial x} \tag{A7}$$

By solving Equations (A5) and (A7), the general solutions of $p(x)$ and $u(x)$ were obtained as:

$$p(x) = P_a e^{-j\beta x} + P_b e^{j\beta x} \tag{A8}$$

$$u(x) = \frac{1}{Z_0} (P_a e^{j\beta x} - P_b e^{j\beta x}) \tag{A9}$$

where

$$Z_0 = \sqrt{\rho\kappa} \tag{A10}$$

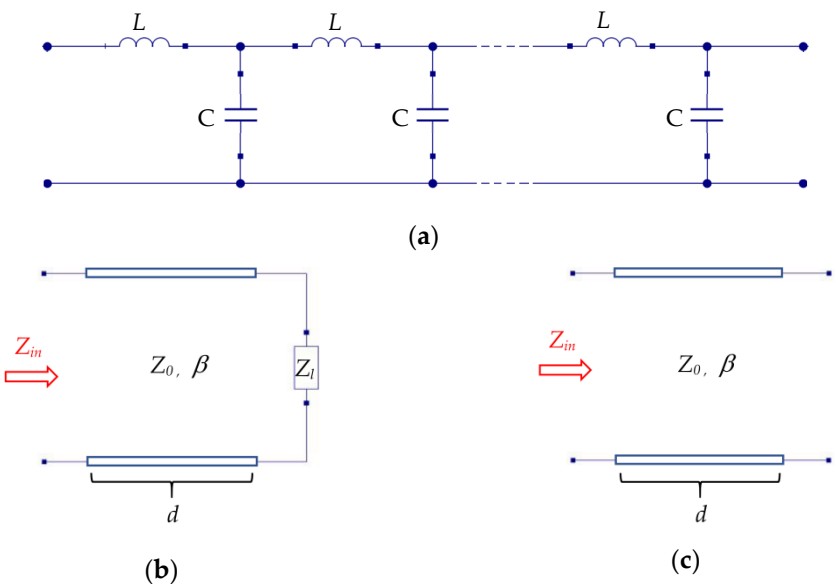

**Figure A1.** (**a**) Ladder network of a transmission line, composed of series inductors $L$ and shunt capacitors $C$. (**b**) $Z_l$–terminated transmission line. (**c**) Open–terminated transmission line.

On the other hand, an equivalent circuit of a transmission line (TL) was expressed by a ladder network composed of series inductors and shunt capacitors, as shown in Figure A1a [14]. When the capacitance and inductance per unit length of the transmission line were defined by $C'$ and $L'$, the telegraph equation of a lossless transmission line was expressed as:

$$\frac{\partial^2 v(x)}{\partial x^2} + \beta^2 v(x) = 0 \tag{A11}$$

where

$$\beta = \omega\sqrt{L'C'} \tag{A12}$$

In addition, since the relation between the voltage $v(x)$ and current $i(x)$ were given by

$$\frac{\partial v(x)}{\partial x} = -j\omega L' i(x) \tag{A13}$$

the general solutions of $v(x)$ and $i(x)$ were given as

$$v(x) = V_a e^{-j\beta x} + V_b e^{j\beta x} \tag{A14}$$

$$i(x) = \frac{1}{Z_0}(V_a e^{j\beta x} - V_b e^{j\beta x}) \tag{A15}$$

where

$$Z_0 = \sqrt{\frac{L'}{C'}} \tag{A16}$$

When we compare Equations (A5) and (A11), (A8) and (A14), (A9) and (A15), (A10) and (A16), and (A6) and (A12), we can see that the equations for the sound waves traveling in $x$ direction were written in the same form as those for electric signals traveling on a transmission line; namely, the propagation of the sound waves could be expressed by the transmission line based equivalent circuits by associating variables and medium constants as follows.

$$p(x) \leftrightarrow v(x) \tag{A17}$$

$$u(x) \leftrightarrow i(x) \tag{A18}$$

$$\rho \leftrightarrow L' \tag{A19}$$

$$\kappa \leftrightarrow 1/C' \tag{A20}$$

Based on the transmission line theory, the input impedance $Z_{in}$ of the $Z_l$–terminated transmission line with length $d$, shown in Figure A1b, was written as [14]:

$$Z_{in} = \frac{Z_l + jZ_0 \tan\beta d}{Z_0 + jZ_l \tan\beta d}Z_0 \tag{A21}$$

The perfect reflection with the maximum voltage and zero current occurred at an open–terminated transmission line, as presented in Figure A1c. By setting $Z_l \to \infty$ in Equation (A21), the input impedance $Z_{in}$ could be derived as [14]:

$$Z_{in} = -jZ_0 \cot\beta d \tag{A22}$$

This corresponds to the case when the cochlea–based symmetric model in Figure 6 was excited by the even mode sound wave. Since the cochlear apex reflected the even mode sound waves perfectly with the maximum pressure and zero velocity, as shown in Figure 7b, the input impedance of the even mode, $Z_{even}$, could be determined as follows by analogy from the input impedance of the open–terminated transmission line in Equation (A22).

$$Z_{even} = -jZ_0 \cot\beta L_c \tag{A23}$$

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
