# Peer review of "Contribution of Even/Odd Sound Wave Modes in Human Cochlear Model on Excitation of Traveling Waves and Determination of Cochlear Input Impedance"

_acoustics, doi:10.3390/acoustics4010011_

Round 1

Reviewer 1 Report

The paper is well-written. Authors could discuss more on advantages of using time domain or frequency domain methods.

Author Response

Dear Mr./Ms. Reviewer 1

Thank you very much for spending your valuable time for the review.

I attach the response as an attachment.

Could you please see it.

Thank you very much again and I wish you my best regards,

Yasushi Horii

Reviewer 2 Report

To be honest, it is very hard to follow what have been done in this study because of the poor English writing and there are too many parameters and very little background information is presented for the model used. This paper needs extensive English editing in order for further submission for publication. There are too many errors in English grammar and word usages.

Introduction:  The paper is mainly about the contribution of even and odd sound wave modes to the excitation of traveling waves, however the introduction section does not present the rationale and give a thorough literature review. Most of the references cited were published before 2014 including some books. Any study should be based on some previous research work and I don't see in the references.

It is really difficult for general readers to understand the model used. The paper should focus on some important changes made.

Fig. 2: It is not clear how the figure in panel (f) was generated and please give more details.

The current version of the paper is not acceptable for consideration of publication. I cannot even judge the validity of the study and its results because of its writing.

Author Response

Dear Mr./Ms. Reviewer 2

Thank you very much for spending your valuable time for the review.

I attach the response as an attachment.

Could you please see it.

Thank you very much again and I wish you my best regards,

Yasushi Horii

Round 2

Reviewer 2 Report

I am happy with the revision. My concerns have been addressed.